# BNIP3L-Dependent Mitophagy Promotes HBx-Induced Cancer Stemness of Hepatocellular Carcinoma Cells via Glycolysis Metabolism Reprogramming

**DOI:** 10.3390/cancers12030655

**Published:** 2020-03-11

**Authors:** Yuan-Yuan Chen, Wei-Hua Wang, Lin Che, You Lan, Li-Yin Zhang, Deng-Lin Zhan, Zi-Yan Huang, Zhong-Ning Lin, Yu-Chun Lin

**Affiliations:** 1State Key Laboratory of Molecular Vaccinology and Molecular Diagnostics, School of Public Health, Xiamen University, Xiamen 361102, China; 32620161150504@STU.XMU.EDU.CN (Y.-Y.C.); 32620181150891@STU.XMU.EDU.CN (W.-H.W.); 32620170154849@STU.XMU.EDU.CN (L.C.); 32620191150617@STU.XMU.EDU.CN (Y.L.); 32620161150505@STU.XMU.EDU.CN (L.-Y.Z.); 32620171150391@STU.XMU.EDU.CN (D.-L.Z.); 32620162200395@STU.XMU.EDU.CN (Z.-Y.H.); 2National Institute of Environmental Health, Chinese Center for Disease Control and Prevention, Beijing 100021, China

**Keywords:** hepatitis B virus x protein, liver cancer stem cells, BNIP3L, mitophagy, glycolysis metabolism reprogramming

## Abstract

Hepatitis B virus (HBV) is one of predisposing factors for hepatocellular carcinoma (HCC). The role of HBV x protein (HBx) in mediating the induction and maintenance of cancer stemness during HBV-related HCC attracts considerable attention, but the exact mechanism has not been clearly elucidated. Here, ABCG2-dependent stem-like side population (SP) cells, which are thought to be liver cancer stem cells (LCSCs), were present in HCC cells, and the fraction of this subset was increased in HBx-expressing HCC cells. In addition, glycolysis was upregulated in LCSCs and HBx-expressing HCC cells, and intervention of glycolysis attenuated cancer stem-like phenotypes. Mitochondria play an important role in the maintenance of energy homeostasis, BNIP3L-dependent mitophagy was also activated in LCSCs and HBx-expressing HCC cells, which triggered a metabolic shift toward glycolysis. In summary, we proposed a positive feedback loop, in which HBx induced BNIP3L-dependent mitophagy which upregulated glycolytic metabolism, increasing cancer stemness of HCC cells in vivo and in vitro. BNIP3L might be a potential therapeutic target for intervention of LCSCs-associated HCC. Anti-HBx, a monoclonal antibody targeting intracellular HBx, had the potential to delay the progression of HBV infection related-HCC.

## 1. Introduction

Hepatocellular carcinoma (HCC) is one of the most common malignant tumors with the high mortality rate, in which hepatitis B virus (HBV) infection is regarded as one of major risk factors [1]. HBV x protein (HBx), encoded by HBV X gene for HBV replication, which plays an important role in the development and progression of HCC via regulating a series of biological processes of the host hepatocytes, such as gene transcription, cell cycle, proliferation, and survival [2]. Recently, emerging studies suggested that HBx promoted cancer stem cells (CSCs) generation in the pathogenesis of HCC [3]. For example, HBx derived alpha fetoprotein (AFP) expression to activate phosphatidylinositide 3-kinases (PI3K)/protein kinase B (AKT) signal pathway to stimulate normal human liver cells to generate hepatic CSCs, thereby accelerating the malignant transformation of hepatic cells [4]. CSCs are a small subset of heterogeneous cell populations which have phenotypes of self-renewal, differentiation, metastasis, and recurrence [5]. Activation of the stemness-related octamer-binding transcription factor 4 (*OCT4*) gene contributed to cell migration, drug resistance, and poor prognosis in HBV-related HCC [6,7]. HBx activated stemness-related proteins, including β-catenin, epithelial cell adhesion molecule (EpCAM), Klf-4, Nanog, and Oct4 in vivo and in vitro [4]. Therefore, exploring the molecular mechanisms by which HBx promoted CSCs generation would be useful to identify a novel interventional target.

HBV infection-triggered alteration of mitochondrial metabolism and mitochondrial quality control (MQC) has recently become a hot topic in cancer research [8]. Mitophagy, one of the critical component of MQC, is a selective degradation process that specifically targets the impaired mitochondria to autophagosomes for maintaining of mitochondria homeostasis. In recent years, the molecular mechanism of mitophagy has been extensively studied. Mitophagy regulatory pathways were classified as PTEN-induced kinase 1 (PINK1)/Parkin-dependent signaling, mitochondrial receptors–dependent signaling such as B-cell lymphoma 2/adenovirus E1B interacting 19 kDa-interacting protein 3 (BNIP3), BNIP3-like (BNIP3L), and FUN14 domain containing 1 (FUNDC1), and lipid-mediated signaling [9,10]. PINK1/Parkin-dependent mitophagy might play a double-faceted role in hepatic cancer cells depending on the different cellular context [11]. On the one hand, HBx induced aberrant mitochondria dynamics and promoted PINK1/Parkin-dependent mitophagy, to promote cell survival and possible viral persistence [12]. On the other hand, it was suggested that thyroid hormone eliminated HBx-targeting mitochondria via PINK1/Parkin pathway in hepatocytes, and consequently prevented HBx-induced HCC [13]. It was reported that HBx directly interacted with the basic helix–loop–helix (bHLH) domain of hypoxia inducible factor-1α (HIF-1α) to increase its transcriptional activity and protein level [14]. Interestingly, the expression of HIF-1α target gene *BNIP3L* has been reported to cause mitochondrial dysfunction and cell death in breast tumors [15,16]. BNIP3L at the outer mitochondrial membrane interacts with the processed microtubule-associated protein light chain 3 (LC3) at phagophore membranes to promote the occurrence of mitophagy. It was considered to be important for mitochondrial clearance during reticulocyte maturation, as well as mitophagy is important for the stemness maintenance in an energy-dependent manner [17,18]. Importantly, mitophagy acts as a key mechanism for developing and maintaining stemness. During chemotherapy, BNIP3L-dependent mitophagy was activated to clear the damaged mitochondria and maintain cell survival in colorectal CSCs [19]. However, whether HBx could induce BNIP3L-dependent mitophagy in the progression of HBV-related HCC remains to be elucidated. Therefore, more detailed experimental investigation underlying the role of mitophagy in the acquisition and maintenance of cancer stemness in HBV-related HCC is worthy of further studying.

Besides, mitophagy regulates the mitochondrial dysfunction that can affect the metabolic reprogramming [20]. In 1930, Otto Warburg, for the first time, suggested that cancer cells with mitochondrial defects and malfunction preferentially underwent glycolysis instead of oxidative phosphorylation (OXPHOS), even in the presence of oxygen [21]. As we known, the production of adenosine triphosphate (ATP) is much more efficient through OXPHOS than glycolysis, so the mild respiratory dysfunction would require a substantial increase of glycolysis to maintain the energy balance [22]. This reprogramming of energy metabolism is one of the hallmarks of cancer cells which require sufficient ATP to supply for their active metabolism and proliferation. The expression of key rate-limiting enzymes, such as glucose transporter 1 (GLUT1), hexokinases (HKs), glyceraldehyde-3-phosphate dehydrogenase (GAPDH), pyruvate kinase, and lactate dehydrogenase (LDHA), were enhanced, and promoted glycolysis of hepatocytes during HCC progression [23]. Studies had shown that HBx was closely related to cellular metabolism. Liu had found that HBx can upregulate glucose-6-phosphate dehydrogenase (G6PD) via the activation of p62-Nrf2-keap1 signaling axis, promoting the pentose phosphate pathway [24]. Besides, HBx increased aberrant glycosylated apolipoprotein B (apoB) to inhibit the secretion of apoB, and then promoted intracellular lipid accumulation [25]. HBx expression also upregulate the transcriptional activity of the sterol regulatory element binding protein-1a (SREBP-1a) [26]. Using nuclear magnetic resonance-based metabolomics methods, it was found that HBx initially induced cellular DNA damage, then disrupted cellular nucleic acid metabolism and prevented DNA repair, inducing HCC [27]. However, there was yet a limited understanding whether HBx can remodel glucose metabolism and what functions and mechanism by which remodeling of glucose metabolism involves in promoting the stemness of HBx-expressing HCC cells.

There are currently 350 million HBV carriers worldwide. The main drugs used for the treatment of HBV infection are nucleoside (acid) analogues and interferon, while they cannot eradicate the virus or completely block the development of hepatocarcinogenesis [28]. HBx is a multifunctional protein, and plays multiple roles in the development of HBV-associated hepatocarcinogenesis [2]. Therefore, HBx is a potential target for therapeutic intervention against HBV infection. Due to the lack of crystal structure of the full-length HBx protein, there is a lack of effective interventions. Zhang recently developed a monoclonal antibody (mcAb), which could specifically target to the intracellular HBx-expressing intervention (anti-HBx) [29]. However, its role in the interfering with HBx-induced cancer stemness remains to be elucidated.

In this study, we hypothesized that HBx promoted the cancer stemness of HCC cells via increasing mitophagy-mediated glycolysis metabolism reprogramming. Multiple HBx-expressing cell models were established, while side population (SP) of ATP-binding cassette sub-family G member 2 (ABCG2) positive subset, or sphere-forming cells with stem-like phenotypes were measured. In the studies of mechanism, we proposed a positive feedback loop that HBx upregulated glycolytic metabolism reprogramming through BNIP3L-dependent mitophagy mediated by HIF-1α transactivation, and consequently enhanced the liver cancer stemness phenotypes. Our research provided a novel mechanism for the stemness of hepatic cancer cells conferred by HBx and raised BNIP3L as a possible therapeutic target for liver cancer stem cells (LCSCs)-associated HCC. Moreover, current results indicated that anti-HBx could reduce the HBx-induced hepatocarcinogenesis, and had the potential to delay the progression of HBV infection related-HCC.

## 2. Results

### 2.1. HBx Promoted HCC Cells Xenograft Tumors Growth via Upregulated Glycolytic Metabolism In Vivo

To evaluate the effect of HBx-expressing on tumor growth, in vivo was conducted using mouse subcutaneous xenograft model. Tumor xenografts derived from HBx-expressing Huh7 and MHCC-97H cells exhibited the tumors with rapider growth, bigger volumes, and higher weights than those from empty vector (pcDNA3.1-HA plasmid)-transduced cells (Figure 1A–C). Compared with control xenograft tumors, the mRNA levels of cancer stemness-related genes, including *ABCG2*, *BMI1*, *NANOG*, *KLF4*, and *OCT4* were increased and the protein expression of ABCG2, Oct4, Bmi-1, and CD44 were also increased in HBx-expressing xenograft tumors (Figure 1D,E). Immunohistochemistry (IHC) analysis also confirmed the above results, cancer stemness-related proteins were upregulated in HBx-harbored xenograft tumors (Figure 1F). Additionally, HBx-expressing decreased the content of ATP, and increased the content of lactic acid in xenograft tumors (Figure 1G,H). The mRNA levels of *SLC2A1*, *HK2*, *PFKL*, and *LDHA* also increased in HBx-expressing group, compared with those in control group (Figure 1I). Altogether, HBx-expressing promoted cancer stemness phenotype and glycolysis metabolism reprogramming in HCC xenograft tumors.

### 2.2. HBx Promoted Cancer Stemness Phenotype of HCC Cells

Based on results of SP cell ratio detected by FCM, we confirmed that Huh7 and MHCC-97H cells contained 1.25% and 16.8% SP cells, and cancer stem-like phenotypes were presented in two HCC cells (Appendix A). To further characterize stemness features of HCC cells, Huh7 cells were cultured in the ultra-low attachment plate for two weeks, while resulted in the enrichment of LCSCs by using sphere-formation assay. WB and qRT-PCR analysis showed that the levels of cancer stemness-related genes and proteins were higher in CSCs than in parental Huh7 cells (Appendix A). Furthermore, to explore the role of HBx-expressing in cancer stemness, two HCC cell lines were transiently transfected with pcDNA3.1-HBx for 8 h and restored for another 48 h to establish the HBx-expressing models. In Figure 2A,B, it was shown that the mRNA levels of cancer stemness-related genes, including *ABCG2*, *BMI1*, *NANOG*, *KLF4*, and *OCT4* were increased, and the protein expression levels of ABCG2, Oct4, Bmi-1, and CD44 were also increased in HBx-expressing HCC cells compared with pcDNA3.1 transfected cells. Moreover, the SP cells ratios were increased in HBx-expressing HCC cells. Ratios of SP cells increased from 1.43% and 14.8% of the total Huh7 and MHCC-97H cells in the control groups to 1.73% and 24.9% in the HBx-expressing groups, respectively (Figure 2C). In parallel to the increased SP cells ratios, the ability of anchorage-independent growth and the number of colony formation were increased in HBx-expressing HCC cells (Figure 2D,E). To further verify the important role of HBx in the cancer stemness of HCC, two HCC cell lines were transiently transfected with pGEM-HBV or pGEM-HBV X null plasmids to establish a HBV or HBV X null expressing model. Compared with HBV X null expressing groups, the levels of cancer stemness-related proteins were increased in pGEM-HBV transfecting groups, which were transfected a 1.3-fold-overlength genome of HBV expression plasmid (Figure 2F). In general, HBx promoted cancer stemness phenotypes in HBV-related HBx-expressing HCC cell lines.

### 2.3. Glycolytic Metabolism Was Reprogrammed in the HBx-Expressing HCC Cells and LCSCs

To clarify the effect of HBx-expressing on glycolytic metabolism and the metabolic pattern of LCSCs in vitro, LCSCs were enriched by sphere-formation assay. As shown in Figure 3A–C, we found that the glucose uptake and the secretion of extracellular lactic acid were increased, while the content of intracellular ATP was decreased in sphere-formed LCSCs, compared with parental Huh7 cells by FCM analysis. In addition, the mRNA levels of glycolysis-related genes, including *SLC2A1*, *HK2*, *PFKL*, and *LDHA,* were significantly higher, whereas the OXPHOS-related genes mRNA levels of *CytB*, *ATP6*, and *ATP8* were lower in sphere-formed LCSCs than those in parental Huh7 cells (Figure 3D,E). Our results suggested that glycolytic metabolism was reprogrammed in LCSCs. Next, the effect of HBx-expressing on glycolytic metabolism in HCC cells was examined. It was found that the ability of glucose uptake, and the secretion of lactic acid were increased, while the content of ATP was decreased in two HBx-expressing HCC cell lines (Figure 3F–H; Appendix A). In addition, the mRNA levels of glycolysis-related genes were significantly higher (Figure 3I; Appendix A), whereas the mRNA levels of OXPHOS-related genes were significantly lower in two HBx-expressing HCC cells than that in two HCC cell lines (Figure 3J; Appendix A). All of these results showed that LCSCs were in favor of glycolytic metabolism, and HBx-expressing triggered a metabolic shift toward glycolysis in HCC cell lines.

### 2.4. Glycolysis Metabolism Reprogramming Regulated Cancer Stemness of HCC Cells and LCSCs

To investigate the role of glycolysis in cancer stemness induced by HBx-expressing, STF-31 and 2-Deoxy-D-glucose (2-DG) were used to inhibit glycolytic metabolism. STF-31 was used to inhibit glucose transport, and 2-DG was used to inhibit the production of glucose-6-phosphate from glucose. MHCC-97H cells with a high proportion of SP cells were chosen for the intervention. The levels of cancer stemness-related genes (*ABCG2*, *BMI1*, *NANOG*, *KLF4*, and *OCT4*) mRNA and proteins (ABCG2, Oct4, Bmi-1, and CD44) expression were downregulated with treatment of STF-31 in HBx-expressing MHCC-97H cells (Figure 4A,B). Consistent with STF-31 treatment, 2-DG intervention also reduced the levels of cancer stemness-related genes mRNA and proteins expression induced by HBx-expressing (Figure 4C,D). Furthermore, STF-31 and 2-DG were used to intervene cancer stemness phenotypes in sphere-formed LCSCs of Huh7 cells. It was found that the levels of cancer stemness-related proteins expression were inhibited in sphere-formed LCSCs (Figure 4E,F). Then, we analyzed the SP cells ratio in MHCC-97H cells treated with STF-31 or 2-DG. The proportion of SP cells decreased from 20.25% to 12.19% in HBx-expressing MHCC-97H cells treated with STF-31. Consistent with the STF-31 treatment, the proportion of SP cells decreased from 19.65% to 15.45% in HBx-expressing MHCC-97H cells treated with 2-DG (Figure 4G). Taken together, these results indicated that glycolytic metabolism would play an important role in maintaining the liver cancer stemness induced by HBx-expressing.

### 2.5. BNIP3L-Dependent Mitophagy Was Induced in HBx-Expressing HCC Cells and LCSCs

Mitochondrion is an important energizing organelle and involves in the regulation of multiple signaling in cells. Mitophagy is one of the ways to maintain mitochondrial homeostasis, and also plays an important role in maintaining cancer stemness in HepG2 cells [30]. The ultra-structures of sphere-formed LCSCs and parental Huh7 cells were analyzed by TEM. It was found that ovoid structure, low cristae density, plentiful autophagosomes, and distinctive perinuclear localization of mitochondria were observed in sphere-formed LCSCs of Huh7, while rare autophagosomes and abundant mature mitochondria were observed in parental Huh7 cells (Figure 5A). The IF analysis showed that the co-localization of BNIP3L, LC3B, and mitochondria was increased in sphere-formed LCSCs (Figure 5B). Moreover, the expression levels of BNIP3L-dependent mitophagy-related proteins, including HIF-1α, Beclin1, BNIP3L, and LC3B, were upregulated in LCSCs (Figure 5C). These results showed that LCSCs had a high level of BNIP3L-dependent mitophagy.

To clarify whether HBx can induce BNIP3L-dependent mitophagy, autophagic flux was detected by tandem fluorescent-tagged LC3 (mTagRFP-mWasabi-LC3) in HBx-expressing Huh7 cells [31]. As shown in Figure 5D, compared with Huh7 cells, it was found that the formation of autophagosomes and autophagic flux were observed, and the quantity of autophagosomes was also increased in HBx-expressing Huh7 cells (Figure 5E). It had been reported that the HIF-1α transcriptionally regulates the expression of BNIP3L for its dependent mitophagy. Compared with Huh7 cells, it was also found that the expression of HIF-1α and BNIP3L-dependent mitophagy-related proteins in whole-cell lysate and mitochondrial fractions were increased in HBx-expressing Huh7 cells (Appendix A, Figure 5F,H). The fluorescence puncta of BNIP3L and LC3B, and the co-localization of BNIP3L, LC3B, and mitochondria was also increased (Figure 5G). Consistent with in vitro, the expression levels of BNIP3L-dependent mitophagy-related proteins were also increased in HBx-expressing Huh7 xenograft tumors (Appendix A). These results showed that HBx-expressing induced BNIP3L-dependent mitophagy via the accelerated autophagic flux in HCC cells.

### 2.6. BNIP3L-Dependent Mitophagy Regulated Cancer Stemness of the HBx-Expressing HCC Cells

To clarify the role of BNIP3L-dependent mitophagy in cancer stemness caused by HBx-expressing, we selected MHCC-97H cells with a high proportion of SP cells to construct a mitophagy inhibition model by using si*BNIP3L*. As shown in Figure 6A, the fluorescence puncta of BNIP3L and LC3B, and the co-localization of BNIP3L, LC3B, and mitochondria induced by HBx-expressing was relieved by si*BNIP3L* transfection. Consistently, the expression of BNIP3L-dependent mitophagy-related mitochondrial proteins induced by HBx-expressing were also decreased by si*BNIP3L* transfection (Figure 6B). Moreover, the cancer stemness-related indicators in HBx-expressing MHCC-97H cells treated with siNC or si*BNIP3L* transfection were determined. The mRNA levels of cancer stemness-related genes and the expression levels of cancer stemness-related proteins were inhibited by si*BNIP3L* transfection in HBx-expressing MHCC-97H cells (Figure 6C,D). Similarly, after si*BNIP3L* transfection, the number of colony formation was also decreased (Figure 6E). Consistent with the above results, the proportion of SP cells decreased from 27.09% to 21.56% in HBx-expressing MHCC-97H cells transfected with si*BNIP3L* (Figure 6F).

Carbonyl cyanide m-chlorophenylhydrazone (CCCP), an agonist of mitophagy, was also used to induce BNIP3L-denpendent mitophagy [32]. Huh7 cells with a low proportion of SP cells were selected to construct the mitophagy positive model by using CCCP treatment. IF analysis revealed that the fluorescence puncta of BNIP3L and LC3B were increased and the co-localization of BNIP3L and LC3B in mitochondria was also increased in HBx-expressing or CCCP treated Huh7 cells (Appendix A). The expression of BNIP3L-dependent mitophagy-related mitochondrial proteins was increased in HBx-expressing or CCCP treated Huh7 cells (Appendix A). Besides, both the levels of cancer stemness-related genes and the expression levels of cancer stemness-related proteins were increased in HBx-expressing or CCCP treated Huh7 cells (Appendix A). Compared with Huh7 cells, the number of colony formation was increased in HBx-expressing or CCCP treated Huh7 cells (Appendix A). After treated with CCCP, the proportion of SP cells increased from 1.57% to 2.54% in Huh7 cells (Appendix A). These results indicated that BNIP3L-dependent mitophagy played an important role in maintaining cancer stemness in HBx-expressing HCC cells.

### 2.7. BNIP3L-Dependent Mitophagy Induced by HBx-Expressing Reprogrammed the Glycolytic Metabolism

To investigate the relationship between BNIP3L-dependent mitophagy and glycolysis metabolism reprogramming in HCC cells, mitophagy intervention models were established. Huh7 cells with low proportion of SP cells were selected to construct a mitophagy positive model by the induction with CCCP, while MHCC-97H cells with high proportion of SP cells were chosen to construct a mitophagy suppression model by si*BNIP3L* transfection. Compared with Huh7 cells, the ability of glucose uptake and the secretion of extracellular lactic acid were increased, while the content of intracellular ATP was decreased in HBx-expressing or CCCP treated Huh7 cells (Figure 7A–C). In addition, the mRNA levels of glycolysis-related genes (*SLC2A1*, *HK2*, *PFKL*, and *LDHA*) were significantly higher in HBx-expressing or CCCP treated Huh7 cells than that in Huh7 cells (Figure 7D). Contrast to the CCCP treatment group, glycolytic metabolism was inhibited in HBx-expressing MHCC-97H cells transfected with si*BNIP3L*. Compared with HBx-expressing MHCC-97H cells, the ability of glucose uptake and the secretion of extracellular lactic acid were inhibited, while the content of intracellular ATP was increased in HBx-expressing MHCC-97H cells transfected with si*BNIP3L* (Figure 7E–G). In addition, the mRNA levels of glycolysis-related genes, including *SLC2A1*, *HK2*, *PFKL*, and *LDHA* were significantly lower in HBx-expressing MHCC-97H cells transfected with si*BNIP3L* (Figure 7H). These results showed that BNIP3L-dependent mitophagy promoted the glycolysis metabolism reprogramming induced by HBx-expressing in HCC cells.

### 2.8. Anti-HBx Targeting Intervention to Intracellular HBx Inhibited the Hepatocarcinogenesis Associated with BNIP3L-Dependent Mitophagy

To confirm the mechanism of mitophagy in regulating glycolysis metabolism reprogramming mediated stemness phenotypes in clinical cohort samples, a GEO database (GSE83148) from chronic HBV-infected patients was selected. In Figure 8A, it was shown that compared with normal liver tissues, the relative mRNA levels of cancer stemness-related genes, such as *BMI1*, *CD44*, *NANOG*, and *POU5F1*, were upregulated in the chronic HBV-infected liver tissues. Next, the chronic HBV-infected liver tissues exhibited relatively high mRNA levels of glycolysis-related genes, including *HK2*, *PFKL*, *PKM2*, and *LDHA* (Figure 8B). Then, analysis of GEO database (GSE83148) also showed that the relative expression level of *MAP1LC3B* was increased in the chronic HBV-infected liver tissues (Figure 8C). Moreover, the relative expression level of *MAP1LC3B* was positively correlated with that of *HK2* (*r* = 0.431, *P* < 0.05), *PFKL* (*r* = 0.457, *P* < 0.05), *PKM2* (*r* = 0.622, *P* < 0.05), and *LDHA* (*r* = 0.24, *P* < 0.05) (Figure 8D). Chronic HBV-infected cohort data indicated that HBV-infection may associate with the induction of cancer stemness phenotype, glycolytic metabolism, and mitophagy in liver, where mitophagy would positively correlate with glycolytic metabolism in hepatocytes.

HBx plays multiple roles in the development of HBV-associated hepatocarcinogenesis [2]. Therefore, HBx is a potential target for therapeutic intervention against HBV infection. In this study, the novel mcAb specifically target to the intracellular HBx-expressing intervention (anti-HBx) was used as previous report [29]. Compared with HBx-expressing Huh7 cells, the expression of cancer stemness and BNIP3L-dependent mitophagy-related proteins was decreased in HBx-expressing Huh7 cells treated with antagonistic anti-HBx (Figure 8E,F). The glucose uptake was higher in HBx-expressing Huh7 cells than HBx-expressing Huh7 cells treated with anti-HBx (Figure 8G). The content of intracellular ATP was increased (Figure 8H) and the secretion of extracellular lactic acid was decreased (Figure 8I) in HBx-expressing Huh7 cells treated with anti-HBx. In addition, the mRNA levels of *SLC2A1*, *HK2*, *PFKL*, and *LDHA* were significantly lower in HBx-expressing Huh7 cells treated with anti-HBx than that in HBx-expressing Huh7 cells (Figure 8J). These results suggested that anti-HBx could reduce the hepatocarcinogenesis associated with HBx-induced BNIP3L-dependent mitophagy and glycolysis metabolism reprogramming in HCC cells.

## 3. Discussion

CSCs, namely tumor-initiating cells, have both stem cell-like and cancer cell characteristics, including multi-directional differentiation potential, self-renewal, and chemotherapy resistance [33]. LCSCs play an important role in the development of liver cancer [34]. The link between HBx-expressing and CSCs in HBV-related HCC has gradually become a research hotspots. It has been reported that HBx promoted a CD44^+^CD133^+^ CSCs subset in malignant transformed L-02 cells and Huh7 cells, and HBx regulated tumorigenicity, self-renewal, and drug resistance in HCC cells [4,35,36]. In the present study, we selected two different HCC cells with distinct proportion of ABCG2^+^ subset, which Huh7 cells had a low ABCG2^+^ subset ratio and MHCC-97H had a high ABCG2+ subset ratio. Except CD44^+^CD133^+^ subset and OV6^+^ subset, we found that HBx upregulated a subset with positive expression of ABCG2, a drug resistance-associated protein, which had been characterized by high expression of cancer stemness-related proteins, including Oct4, Bmi-1, and CD44. Our previous study found that ABCG2^+^ subset existed in nasopharyngeal carcinoma cells with cancer stemness phenotype [37]. HBx-expressing also upregulated the ability of self-renewal, while the ability of anchorage-independent growth and colony formation were increased in HBx-expressing Huh7 and MHCC-97H cells. Using HBV genome expressing and HBV X knockout model, it was validated the role of HBx-expressing in cancer stem-like phenotypes of HCC cells. In vivo, HBx-expressing promoted xenografted tumor growth and upregulated the expression of cancer stemness-related proteins. Our results indicated that HBx could promote the liver cancer stemness, and its exact mechanism was further studied.

HBx plays an important role in HCC progression via regulating cell cycle, and proliferation [2]. Emerging studies suggested that HBx promoted CSCs generation in the pathogenesis of HCC [3]. However, the regulatory mechanisms that HBx promoted the formation of CSC by inducing mitophagy in the progression of HCC would be elucidated. Mitophagy, selectively degrading excessive or damaged mitochondria, is critical in MQC [38,39,40]. Liu et al. found that mitophagy inhibited the nuclear translocation of p53^Ser392^, and inhibited p53 binding to the *NANOG* promoter to prevent OCT4 and SOX2 transcription factors from increasing the expression of NANOG in HepG2 cells [30]. Defects of PINK1-dependent mitophagy reduced the efficiency of induced pluripotent stem cells (iPSCs) reprogramming in mouse embryonic fibroblasts, with a large number of immature mitochondria, strong spontaneous differentiation and formation of heterogeneous cell populations [41]. In the present study, we found that HBx-expressing induced BNIP3L-dependent mitophagy in vivo and in vitro. We constructed an inducible model of mitophagy in Huh7 cells with a low proportion of SP cells, while constructed an inhibited model of mitophagy in MHCC-97H cells with a high proportion of SP cells. In CCCP-induced mitophagy models, we showed that the liver cancer stemness and the ability of colony formation was accelerated in Huh7 cells. Conversely, intervening the BNIP3L-dependent mitophagy by specific genetic intervention with si*BNIP3L* resulted in the decrease of liver cancer stemness and the ability of colony formation in HBx-expressing MHCC-97H cells. Moreover, Xiang found that SKP/SKO (Sox2, Klf4, Pou5f1/Oct4) stemness genes overexpression induced mouse embryonic fibroblasts (MEFs) reprogramming, BNIP3L-dependent mitophagy decreased the mitochondrial mass and promoted the transformation of MEFs into stem cells [18]. These results further emphasized the role of BNIP3L in cancer stemness maintenance in HBx-expressing HCC cells.

Mitophagy also plays an important role in the maintenance of energy homeostasis. Our study provided a novel mechanism that BNIP3L-dependent mitophagy promoted glycolysis metabolism reprogramming in HBx-expressing HCC cells. Constructing different mitophagy models through mitophagy inducer or specific genetic intervention, we showed that HBx-induced BNIP3L-dependent mitophagy promoted cellular glucose uptake and glycolysis-related gene transcription in HCC cells. It was suggested that BNIP3L-dependent mitophagy promoted the conversion of metabolism from OXPHOS to glycolysis. It was known that metabolism reprogramming has been identified as a hallmark of cancer [42]. Stimulation of glycolysis is critical for the conversion of differentiated cells to a stem-like state and supports the maintenance of stemness of CSCs [43]. CSCs undergo metabolism reprogramming characterized by a reduced mitochondrial activity and a stronger dependence on aerobic glycolysis to adapt to the microenvironment for survival, namely “Warburg effect version 2.0” [44]. Indeed, the increased metabolic switch from OXPHOS to glycolysis had been shown to increase CSC-like phenotypes in HCC [45], breast cancer [46], pancreatic cancer [47], and etc. In addition, it was indicated that radioresistant nasopharyngeal carcinoma CSCs exhibited a greater reliance on glycolysis and that their stem-like phenotypes could be curbed via inhibition of glycolysis [48]. Compared to the parental cells, the upregulated lactic acid production and glycolytic genes were observed in CD133^−^/CD44^+^ mesenchymal glial stem cells [49]. There are still differences in metabolic patterns in different types of cancer stem cells. Wong et al found that CD133^+^ LCSCs had high glycolysis rate and glycolytic capacity [50]. Further, we also found that LCSCs were partial to glycolytic metabolism, and intervening glycolytic metabolism could inhibit the liver cancer stemness. Moreover, HBx triggered a metabolic shift toward glycolysis to increase the cancer stemness of HCC cell lines, while HBx-induced LCSCs was reduced when glycolysis metabolic signaling pathway was inhibited. It was suggested that glycolytic metabolism reprogramming would play an important role in maintaining the LCSCs induced by HBx-expressing in HCC cells.

The metabolic change is one of the important adoptive responses in cancer cells to cope up with continuous requirement of cell survival and proliferation [51]. Viruses enhance aerobic glycolysis upon virus-induced cell transformation, supporting rapid cell proliferation. In our study, HBx-expressing enhanced the cellular glucose uptake, glycolysis-related genes transcription and lactic acid production. Moreover, we analyzed two different GEO databases from viruses infected cohort or viral oncoprotein overexpression cells, including HBV-infected patients liver tissues and normal liver tissues (GSE83148), EBNA-2 overexpression BL41K3 cells (GSD2038). HBV-infected liver tissues and EBNA-2 overexpressing cells exhibited high expression levels of glycolysis-related genes, including *HK2*, *PFKL*, *PKM2*, and *LDHA* (Appendix A). In addition, it is known that multiple viral onco-proteins can also induce cellular metabolic reprogramming. In nasopharyngeal carcinoma cells, latent membrane protein 1 (LMP1) reduced mitochondrial membrane potential, impaired mitochondrial OXPHOS functions, and stimulated glycolysis-related genes to promote aerobic glycolysis [52]. In epithelium, overexpression of human papillomaviruses E2 proteins induced mitochondrial ROS and increased glycolysis through stabilization of HIF-1α [53]. 

In our previous research, protein phosphatase 2A-B55δ subunit enhanced the sensitivity of HCC to cisplatin (cDDP) chemotherapy [54]. In the present study, anti-HBx, the novel mcAb specifically target to the intracellular HBx protein were used. We found that anti-HBx could reduce the liver cancer stemness, BNIP3L-dependent mitophagy, and glycolysis metabolism reprogramming in HBx-expressing HCC cells. Previous studies found that anti-HBx could reduce intracellular HBx via Fc binding receptor TRIM21-mediated protein degradation, and also activated NF-κB, AP-1, and IFN-β, which increased the antiviral status of host cells [29]. These found indicated that anti-HBx may be a new rescuable intervention to block the progression of HBx-induced BNIP3L-dependent mitophagy and glycolysis metabolism reprogramming associated with hepatocarcinogenesis.

## 4. Materials and Methods

### 4.1. Gene Expression Omnibus (GEO) Data Analysis

The public mRNA levels datasets GSE83148 and GDS2038 were screened from GEO database (http://www.ncbi.nlm.nih.gov/geo/). The GDS2038 database samples were derived from Epstein-Barr virus (EBV) negative B-cell lines BL41K3 and EBV nuclear antigen 2 (EBNA-2) overexpressing BL41K3 cells [55]. The GSE83148 database samples were derived from normal liver tissues biopsy and chronically HBV-infected patient liver tissues biopsy [56]. The relative gene expressions of glycolysis-related genes, cancer stemness-related genes, and mitophagy-related gene (*MAP1LC3B* gene for encoding LC3B) in liver tissues were analyzed. Correlation between the expression of *MAP1LC3B* and glycolysis-related metabolic enzymes was evaluated with the Pearson’s correlation analysis.

### 4.2. Xenograft Tumor Study

BALB/c nude mice (5–6 weeks-old) were obtained from the Laboratory Animal Center of Xiamen University (Xiamen, China). All experiments were approved by the Experimental Animal Ethics Committee of Xiamen University (Ethic protocol code: XMULAC20180094). For tumor growth assay, human HCC Huh7 or MHCC-97H cells were transfected with pcDNA3.1-HA or pcDNA3.1-HA-HBx plasmids (a gift from Professor Yuan, Xiamen University) by using Lipofectamine^®^ 2000 (Invitrogen, Carlsbad, CA, USA). Then the cells were selected with 300 μg/mL hygromycin B for two weeks [57]. The mice were randomly allocated to four groups (*n* = 6 for each group) as following, Huh7-pcDNA3.1-HA, Huh7-pcDNA3.1-HA-HBx, MHCC-97H-pcDNA3.1-HA, and MHCC-97H-pcDNA3.1-HA-HBx. A total of 4 × 10^6^ HBx-expressing cells or control cells were suspended in cold phosphate buffered saline (PBS) with matrigel in a 1:1 ratio. The transfected cells were subcutaneously inoculated into the right flank of each mouse, and the tumors were allowed to grow for 24 days. Because of the experiment requires and adhered to the ethical norms of animal experiments, the nude mice were sacrificed immediately after the tumor volume exceeds 4000 mm^3^. All mice were sacrificed and tumors were excised. Tumor width (W), tumor length (L), and body weights were measured every three days. Tumor volume (V) was calculated with the formula: V = (W^2^ × L)/2 [37]. Xenograft tumors were used for subsequent experiments.

### 4.3. Cell Culture

Human hepatocellular carcinoma Huh7 and MHCC-97H cell lines were obtaineded from ATCC and maintained in our laboratory. Cells were cultured in Dulbecco’s Modified Eagle’s medium (DMEM) (Hyclone, Logan, UT, USA) containing 10% fetal bovine serum (FBS) and 1% penicillin-streptomycin (Thermo, Waltham, MA, USA). 

### 4.4. Sphere-Formation Assay

Using StemPro^TM^ Accutase^TM^ cell dissociation reagent digest human Huh7 cells, then cells were resuspended in KnockOut DMEM/F-12 supplemented with 20 ng/mL human EGF, 10 ng/mL human FGF, 2% B27 supplement without vitamin A, and 1% N2 supplement. Huh7 cells (10^4^ cells/well) were cultured in the ultra-low attachment plate (Coring, New York, NY, USA) for two weeks. All these cells were cultured in a humidified incubator with 5% CO_2_ at 37 °C [58]. All the reagents for cell culture, except those as otherwise stated, were obtained from Thermo.

### 4.5. Cell Transfection

Constructing a HBx-expressing cell model, the plasmid pcDNA3.1 and pcDNA3.1-HBx were used, from the National Institute of Diagnostics and Vaccine Development in Infectious Diseases (Xiamen University). For transient overexpression, Huh7 and MHCC-97H cells were transfected with either pcDNA3.1 or pcDNA3.1-HBx (1 μg/mL) by using Lipofectamine^®^ 2000 (Invitrogen).

Constructing a HBV-expressing and its HBx-null cell model, the plasmid pGEM, pGEM-HBV, and pGEM-HBx null was used. For transient overexpression, Huh7 and MHCC-97H cells were transfected with either pGEM, pGEM-HBV, or pGEM-HBx null (1 μg/mL) by using Lipofectamine^®^ 2000 (Invitrogen).

Constructing a HBx-expressing knockdown cell model, the plasmid pTT5 or pTT5-9D11, which was transfected without or with the production of mcAb that targeted intracellular HBx (anti-HBx), was used. Huh7 and HBx-expressing Huh7 cells were transfected with pTT5 or pTT5-9D11 (200 ng/mL) by using Lipofectamine^®^ 2000 (Invitrogen) according to previous research [29].

The plasmid mTagRFP-mWasabi-LC3 was presented by Professor Wang from Capital Medical University (Beijing, China). The mTagRFP-mWasabi-LC3 Plasmid transfections referenced previous research [1]. For transient BNIP3L knockdown, the cells were transfected with 50 nmol/L annealed double-stranded siRNA for BNIP3L (si*BNIP3L*) or a negative control (siNC), (Ribobio, Guangzhou, China), using Lipofectamine^®^ 2000. All experiments were repeated at least three times independently.

### 4.6. Sorting and Analysis of SP Cells and Main Population (MP) Cells

Huh7 and MHCC-97H cells were washed by PBS, digested with 0.25% trypsin-EDTA (Life, Carlsbad, CA, USA) and resuspended in high-glucose DMEM supplemented with 2% FBS at a concentration of 1 × 10^6^ cells/mL. Next, the cells were incubated with 5 μg/mL Hoechst 33342 for 90 min in the dark with mixing every 15 min. Meanwhile for a control, a subset of cells was incubated with 100 μmol/L verapamil for 15 min at 37 °C before the addition of Hoechst 33342. Using cold PBS supplemented with 2% FBS wash the cells twice. Then the cells were analyzed and sorted using a MoFlo Astrios EQS cell Sorter (Beckman, Brea, CA, USA) (355 nm ex, 450/20 and 675 em) as our previous study [37]. Fractions of sorted SP and MP cells were collected for further experiments. The SP cells were a group of low-signal cell populations that have the ability to efflux Hoechst 33342, representing CSCs. The MP cells were defined as the region with high blue and red fluorescence, representing non-CSCs. The ratios of SP and MP to total count cells were calculated.

### 4.7. Anchorage-Independent Growth Assay

Firstly, pretreatment of 6-well plates with 0.8% and 1.2% agarose gel, then 5 × 10^3^/well Huh7 and MHCC-97H cells were plated in 2 mL complete medium, on top of 0.8% of agarose base layer in 6-well plates [59]. The 7, 14, and 21 Days, the shape of formed colonies were observed under the microscope (Nikon, Tokyo, Japan).

### 4.8. Colony Formation Assay

The Huh7 and MHCC-97H cells were digested, resuspended, and 2 mL complete medium containing 1000 cells was added to a 6-well plate. After treatment, cells were cultured for 8–14 days at 37 °C, 5% CO_2_. After most of the cells had formed colonies of >50 cells, and then the supernatant was discarded. Next, the cells were washed 3 times with PBS. The cells were then fixed with 4% paraformaldehyde for 30 min and stained with methylrosanilinium chloride for 20 min [60]. Colonies were analyzed by ImageJ software (Java, Bethesda, MD, USA).

### 4.9. Glucose Transport Assay

Cells were washed by PBS, digested with 0.25% trypsin-EDTA (Life) and resuspended in high-glucose DMEM at a concentration of 5 × 10^5^ cells/mL, then incubated with 50 μmol/L 2-(N-(7-nitrobenz-2-oxa-1,3-diazol-4-yl) amino)-2-deoxyglucose (2-NBDG) (Cayman Chemical, Ann Arbor, MI, USA) for 30 min at 37 °C. Subsequently, cells were washed and resuspended by cold PBS [61]. The ability of glucose uptake was measured (488 nm ex, 530/30 nm em) by using a flow cytometry (FCM) (Beckman, Brea, CA, USA).

### 4.10. ATP Content and Lactic Acid Secretion Assay

Intracellular ATP content and extracellular lactic acid secretion was detected by the ATP Assay Kit (Beyotime, Shanghai, China) and the Lactic acid Assay Kit (Jiangcheng, Nanjing, China) according to the instructions, respectively. The optic density was detected by multifunctional microplate reader (BMG LRBTECH CLARIOstar, Offenburg, Germany).

### 4.11. Transmission Electron Microscope (TEM) Assay

Hepatic CSCs obtained by sphere-formation assay and Huh7 cells were collected and fixed with 2.5% glutaraldehyde buffer at 4 °C for 2 h. Next, cells were post-fixed with 1% osmium tetroxide for 2 h and dehydrated with ethanol and propylene oxide. Then, the cell pellets were embedded in epoxy resin. About 70 nm sections were sliced and stained with lead citrate. Finally, mitophagy and mitochondria morphology were observed by TEM (FEI Tecnai 20, Hillsboro, OR, USA).

### 4.12. Immunofluorescence (IF) Assay and Analysis

Huh7 and MHCC-97H cells were plated in 12 well plates transfected with pcDNA3.1-HBx or si*BNIP3L*. End of treatment, cells were labeled for mitochondria with MitoTracker (Thermo, Waltham, MA, USA) for 30 min at 37 °C. After washing, fixed in 4% paraformaldehyde (pH 7.4) for 30 min, cells were perforated by 0.5% Triton™ X-100 and blocked with 1% bull serum albumin (BSA) for 30 min at room temperature. Cells were incubated with primary antibody (anti-BNIP3L and anti-LC3B) at 4 °C overnight, goat anti-mouse/rabbit IgG fluorescent secondary antibody for 1 h at room temperature. Then cells were visualized on confocal microscope (Leika SP8, Wetzlar, Germany) equipped with ×63 oil objective. The whole process was operated in the dark. Spectral scanning analysis the co-localization of proteins on mitochondria was analyzed by ZEN software (Zeiss, Germany). Quantification of proteins localization on nucleus was analyzed by ImagePro-plus 6.0 (Bethesda, MD, USA). 

### 4.13. Immunohistochemistry (IHC) Assay

The tissue sections from the xenograft tumors in nude mice were deparaffinized in xylene for 20 min and rehydrated with a graded series of ethanol. IHC assay was detected by using the UltraSensitive^TM^ SP Kit (MXB Biotechnologies, Fuzhou, China) according to the instructions. Pictures were taken by an upright microscope (Nikon, Tokyo, Japan). Brown staining indicated immunoreactivity.

### 4.14. Isolation of Mitochondrial-Cytosolic and Nuclear-Cytosolic Fractions

The Nuclear and Cytoplasmic Protein Extraction Kit (Beyotime, Shanghai, China) was applied for nuclear and cytosolic fractions separation according to the manufacturer’s instruction. The mitochondrial and cytosolic fractions were separated by Mitochondria/Cytosol fractionation kit (Enzo Life Science, Farmingdale, Germany) according to the manufacturer’s instruction. The isolated proteins were quantified by the bicinchoninic acid protein assay reagent kit (Beyotime, Shanghai, China) and prepared for western blot. Histone 3, COX IV and actin were used as loading controls for the nucleus, mitochondria and cytoplasm fractions, respectively.

### 4.15. Western Blots (WB)

Cells or liver tissues were lysed for sodium dodecyl sulfate-polyacrylamide gel electrophoresis (SDS-PAGE) in 1× loading buffer or radio immunoprecipitation assay buffer (Beyotime, Shanghai, China) with 1% protease inhibitors and phosphatase inhibitor cocktail. Samples were loaded in SDS lysis buffer (10% SDS, 0.25 M Tris-HCl at pH 6.8, 50% glycerol, 1% β-mercaptoethanol, bromophenol blue), separated by SDS-PAGE and then transferred to polyvinylidene fluoride membranes (Millipore, Bedford, MA, USA). After blocking with 5% skimmed milk for 1 h, the membranes were incubated with the primary antibodies at 4 °C overnight and then with the HRP-labeled secondary antibody for 1 h. Western blot band were visualized with an enhanced chemiluminescence kit (Advansta, Menlo Park, CA, USA). The blot intensities were analyzed by ImageJ software and the relative expression values were calculated.

The primary antibodies for the biotin-labeled secondary antibodies (anti-rabbit IgG and anti-mouse IgG) (dilution 1:5000), anti-BNIP3L (dilution 1:1000), anti-Beclin1(dilution 1:1000), were purchased from Cell Signaling Technology (MA, USA). Anti-HIF-1α antibody was obtained from Novus (Waltham, MA, USA; dilution 1:1000). Anti-actin (dilution 1:10000), and anti-HBx (dilution 1:2000) were obtained from Abcam (Cambridge, MA, USA). Anti-LC3B was obtained from Santa cruze (Shanghai, China; dilution 1:1000). Anti-CD44, anti-Bmi-1, anti-histone 3, and anti-COX IV were obtained from Beyotime (Shanghai, China; dilution 1:1000). Anti-ABCG2 and anti-Oct4 were purchased from Ruiying Biological (Suzhou, China; dilution 1:1000). 

### 4.16. Quantitative Real-Time Polymerase Chain Reaction (qRT-PCR)

Total RNA in the tissues or cells was isolated by TRIzol reagent (TaKaRa Bio, Osaka, Japan). The RNA was reverse transcribed to cDNA by reverse transcriptase (TaKaRa Bio) according to the instructions. qRT-PCR was conducted with SYBR® Premix ExTaqTM II kit (TaKaRa Bio) using a CFX96 Touch^TM^ Detection System (Bio-Rad, Hercules, CA, USA). Cycling conditions were 95 °C for 30 s, 40 cycles of 95 °C for 5 s and 60 °C for 34 s. All primers were described in Appendix A. The expression of target genes was evaluated using the 2^−△△Ct^ relative quantification method, normalized to *ACTB*.

### 4.17. Statistical Analysis

Statistical analyses were conducted using the Statistical Package for Social Sciences (SPSS) version 16.0. All data are shown as mean ± standard deviation (SD). Statistical significance was determined using one-way analysis of variance (ANOVA) and student’s *t*-test. Pearson’s correlation analysis was conducted for variables correlation. *P* < 0.05 was considered to be statistically significant.

## 5. Conclusions

In summary, we proposed a positive feedback loop, in which HBx-expression induced BNIP3L-dependent mitophagy which upregulated glycolytic metabolism reprogramming, increasing cancer stemness of HCC cells in vivo and in vitro (Figure 9). BNIP3L might be a potential therapeutic target for intervention of LCSCs-associated HCC. Anti-HBx, a monoclonal antibody targeting to the intracellular HBx, had the potential to delay the progression of HBV infection related-HCC.

## Figures and Tables

**Figure 1 cancers-12-00655-f001:**
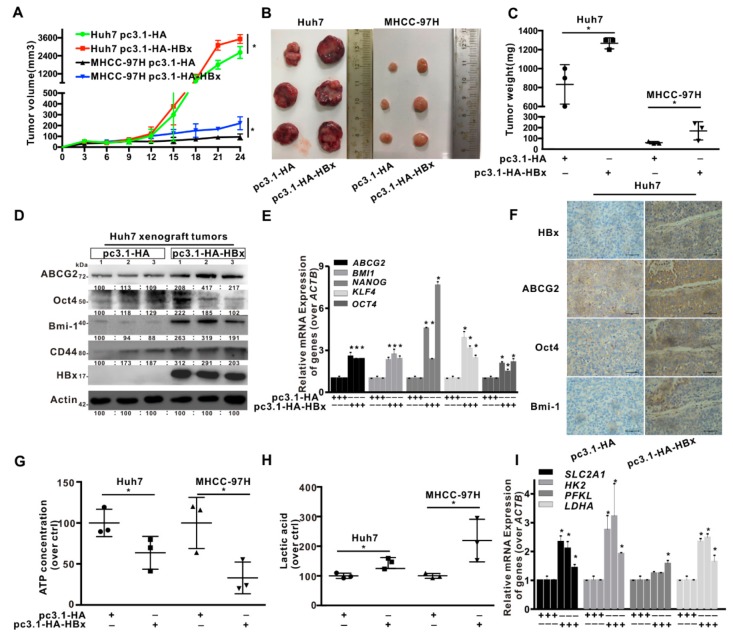
HBV x protein (HBx) promoted the hepatocellular carcinoma (HCC) xenograft tumors growth via the upregulation of glycolytic metabolism in vivo. The HCC xenograft tumors with or without HBx-expressing were formed by Huh7-/MHCC-97H-pcDNA3.1-HA-HBx or Huh7-/MHCC-97H-pcDNA3.1-HA cells in BALB/c nude mice for 24 days. *n* = 6. (**A**) The growth curves of tumors were measured every two days. (**B**) Excised tumors were photographed after mice were sacrificed. (**C**) Tumor weights in the four groups. (**D**) The expression level of the cancer stemness related-proteins in the Huh7 xenograft tumors with or without HBx-expressing. The gray value of band was assessed by image-pro plus 6.0. The relative expression level was showed. (**E**) The mRNA levels of the cancer stemness-related genes in the Huh7 xenograft tumors with or without HBx-expressing. (**F**) Immunohistochemical staining of the cancer stemness-related proteins in the Huh7 xenograft tumors with or without HBx-expressing. Scale bar represents 50 μm. The levels of ATP content (**G**) and lactic acid (**H**) were detected in Huh7 and MHCC-97H xenograft tumors. (**I**) The mRNA levels of glycolysis-related genes in Huh7 xenograft tumors with or without HBx-expressing. The target gene transcription was normalized to *ACTB*. ** P* < 0.05 as compared with pc3.1-HA group. pc3.1-HA: pcDNA3.1-HA transfection without HBx-expressing. pc3.1-HA-HBx: pcDNA3.1-HA-HBx transfection with HBx-expressing.

**Figure 2 cancers-12-00655-f002:**
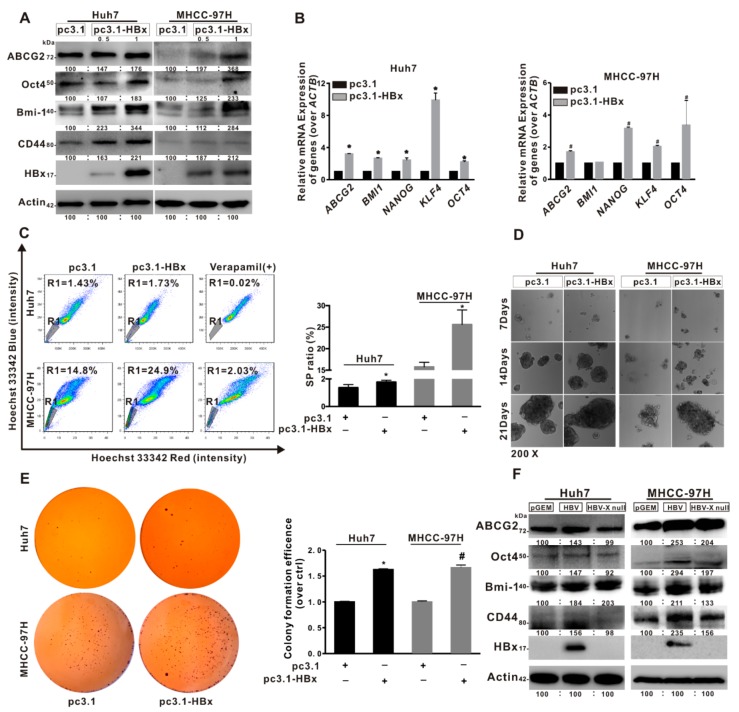
HBx promoted cancer stemness phenotype of the HCC cells. (**A**–**E**) The two HCC cell lines without or with HBx-expressing were transiently transfected with pcDNA3.1 or pcDNA3.1-HBx (0.5, 1 μg/mL) for 8 h and then restored culture for another 48 h. (**A**,**B**) The expression levels of cancer stemness-related proteins (**A**), and the mRNA levels of cancer stemness-related genes (**B**) in two HCC cells without or with HBx-expressing. The target gene transcription was normalized to *ACTB*. (**C**) Percentage of the sorted SP (R1 gate) in two HCC cells without or with HBx-expressing (Left). Quantitative results were shown in bar graph (Right). SP: side population. R1 gate represented SP cells. (**D**,**E**) The self-renewal capacity was analyzed by anchorage-independent growth assay (**D**) and colony formation assay (**E**) in two HCC cells without or with HBx-expressing. Magnification (200×). (**F**) The expression levels of cancer stemness-related proteins in two HCC cells transiently transfected with pGEM, pGEM-HBV, and pGEM-HBV X null plasmids. The gray value of band was assessed by image-pro plus 6.0. The relative expression level was shown. ** P* < 0.05 as compared with Huh7-pc3.1 group. *^#^ P* < 0.05 as compared with MHCC-97H-pc3.1 group. pc3.1: pcDNA3.1 transfection without HBx-expressing. pc3.1-HBx: pcDNA3.1-HBx transfection with HBx-expressing.

**Figure 3 cancers-12-00655-f003:**
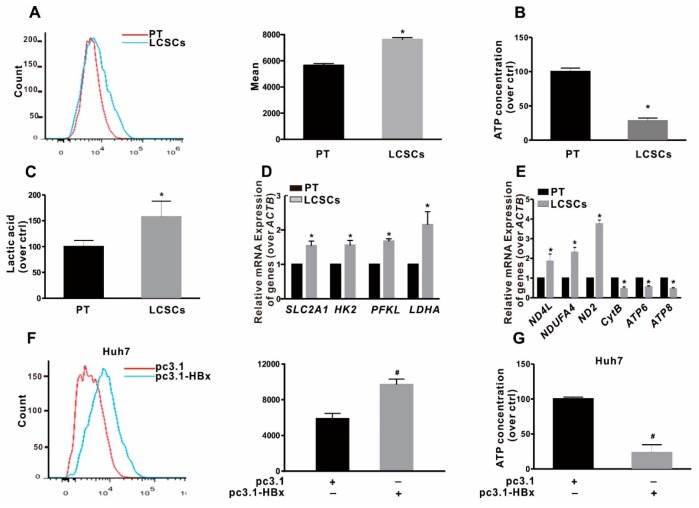
Glycolytic metabolism was reprogrammed in HBx-expressing Huh7 cells and liver cancer stem cells (LCSCs). (**A**–**E**) LCSCs were enriched in Huh7 cells by sphere-formation assay, and normally cultured Huh7 cells served as the parental cells (PT) control group. (**A**) Glucose transport activity was evaluated by Flows cytometry (FCM) (Left). The mean data was shown in bar graph (Right). The levels of intracellular ATP content (**B**), the extracellular lactic acid secretion (**C**), the mRNA levels of glycolysis-related genes (**D**), and the mRNA levels of OXPHOS-related genes (**E**) were detected in PT and LCSCs of Huh7 cells. (**F**–**J**) Huh7 cells without or with HBx-expressing were transiently transfected with pcDNA3.1 or pcDNA3.1-HBx (1 μg/mL) for 8 h, and followed by restored culture for another 48 h. (**F**) Glucose transport activity was evaluated by FCM (Left). The mean data was shown in bar graph (Right). The levels of the intracellular ATP content (**G**), the extracellular lactic acid secretion (**H**), the mRNA levels of glycolysis-related genes (**I**), and the mRNA levels of OXPHOS-related genes (**J**) were detected in Huh7 cells without or with HBx-expressing. The target gene transcription was normalized to *ACTB*. * *P* < 0.05 as compared with PT cells group. ^#^
*P* < 0.05 as compared with pc3.1 group. pc3.1: pcDNA3.1 transfection without HBx-expressing. pc3.1-HBx: pcDNA3.1-HBx transfection with HBx-expressing.

**Figure 4 cancers-12-00655-f004:**
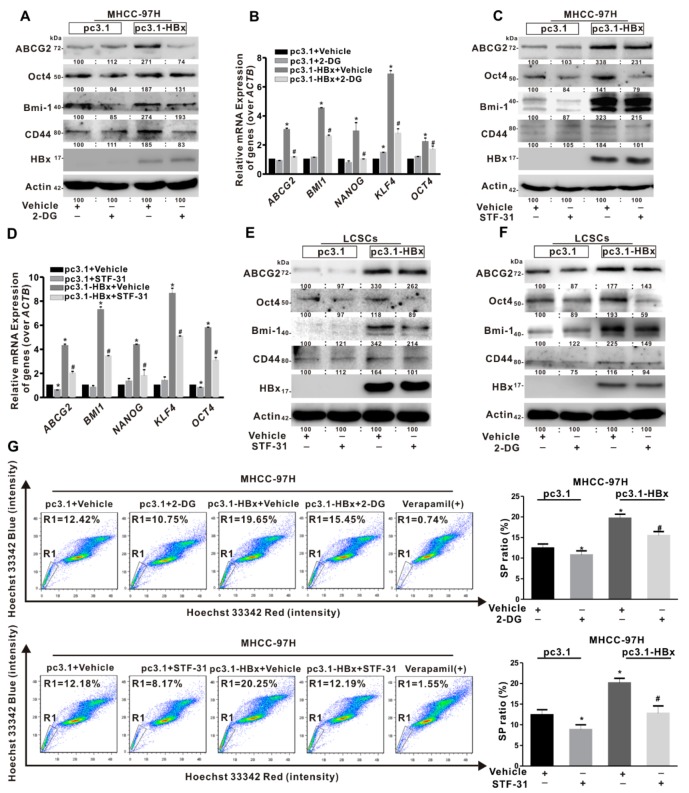
Glycolytic metabolism regulated cancer stemness of HBx-expressing MHCC-97H cells and LCSCs. MHCC-97H cells without or with HBx-expressing were transiently transfected with pcDNA3.1 or pcDNA3.1-HBx (1 μg/mL) for 8 h, followed by treatment with 2-DG (4 μmol/L) or STF-31 (8 μmol/L) or not for another 24 h. (**A**,**B**) The expression levels of cancer stemness-related proteins (**A**) and the mRNA levels of cancer stemness-related genes (**B**) in HBx-expressing MHCC-97H cells treated with 2-DG or not. (**C**,**D**) The expression levels of cancer stemness-related proteins (**C**) and the mRNA levels of cancer stemness-related genes (**D**) in HBx-expressing MHCC-97H cells treated with STF-31 or not. The target gene transcription was normalized to *ACTB*. (**E**,**F**) LCSCs were enriched by sphere-formation assay in MHCC-97H cells. The expression levels of cancer stemness-related proteins in HBx-expressing sphere-formed LCSCs treated with STF-31 (E) or 2-DG (F) or not. The gray value of band was assessed by image-pro plus 6.0. The relative expression level was shown. (**G**) Percentage of the sorted SP cells (R1 gate) in HBx-expressing MHCC-97H treated with 2-DG (Upper) or STF-31 (Lower) or not were detected by FCM (Left). Quantitative results were shown in bar graph (Right). SP: side population. R1 gate represented SP cells. * *P* < 0.05 as compared with pc3.1+Vehicle group, ^#^
*P* < 0.05 as compared with pc3.1-HBx+Vehicle group. pc3.1: pcDNA3.1 transfection without HBx-expressing. pc3.1-HBx: pcDNA3.1-HBx transfection with HBx-expressing.

**Figure 5 cancers-12-00655-f005:**
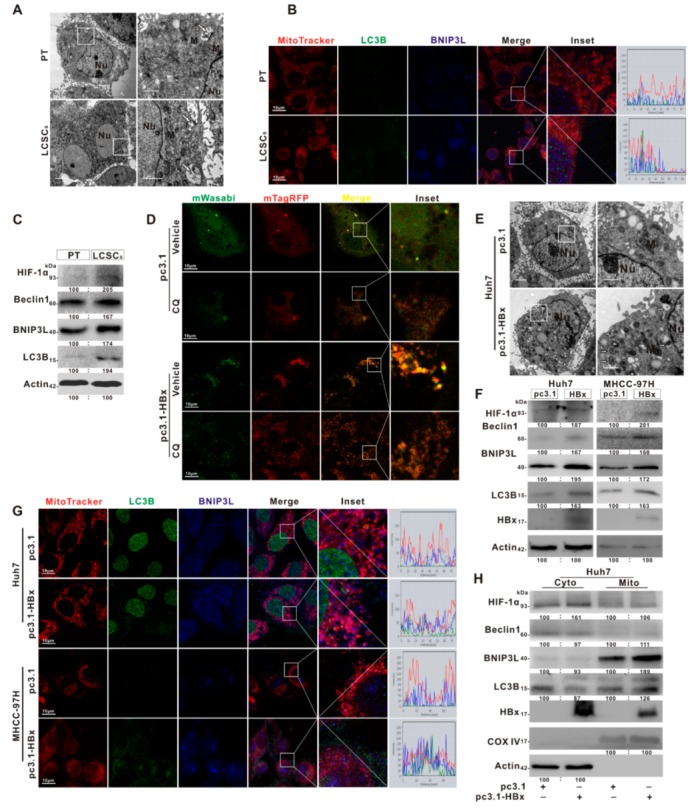
BNIP3L-dependent mitophagy was induced in HBx-expressing HCC cells and LCSCs. (**A**–**C**) LCSCs were enriched in Huh7 cells by sphere-formation assay, and normally cultured Huh7 cells served as the parental cells (PT) control group. (**A**) Mitochondrial ultrastructures were analyzed by TEM. (**B**) Representative images of the immunofluorescence co-staining for MitoTracker (red), BNIP3L (blue), and LC3B (green). Scale bar represents 10 μm. (**C**) The expression levels of BNIP3L-dependent mitophagy-related proteins. (**D**–**H**) Huh7 and MHCC-97H cells without or with HBx-expressing were transiently transfected with pcDNA3.1 or pcDNA3.1-HBx (1 μg/mL). (**D**) Representative fluorescent images of Huh7 and HBx-expressing Huh7 cells were transiently transfected with mTagRFP-mWasabi-LC3 with the pretreatment of chloroquine (CQ, 20 μg/mL) or not. (**E**) Mitochondrial ultrastructures in Huh7 and HBx-expressing Huh7 cells were analyzed by TEM. Scale bar represents 2 μm (Left) or 1 μm (Right). (**F**) The protein expression of BNIP3L-dependent mitophagy in HCC cells and their HBx-expressing cells. (**G**) Representative images of the immunofluorescence co-staining for MitoTracker (red), BNIP3L (blue), and LC3B (green) in HCC cells with or without HBx-expressing. The profiles of representative lines trace the intensities of fluorescence signals. Fluorescence curves with line intensity profile generated by Zen 2012 software were shown. (**H**) The protein expression of BNIP3L-dependent mitophagy in cytoplasmic (Cyto) and mitochondrial (Mito) fractions of Huh7 cells and its HBx-expressing cells. The gray value of band was assessed by image-pro plus 6.0. The relative expression level was shown. pc3.1: pcDNA3.1 transfection without HBx-expressing. pc3.1-HBx: pcDNA3.1-HBx transfection with HBx-expressing.

**Figure 6 cancers-12-00655-f006:**
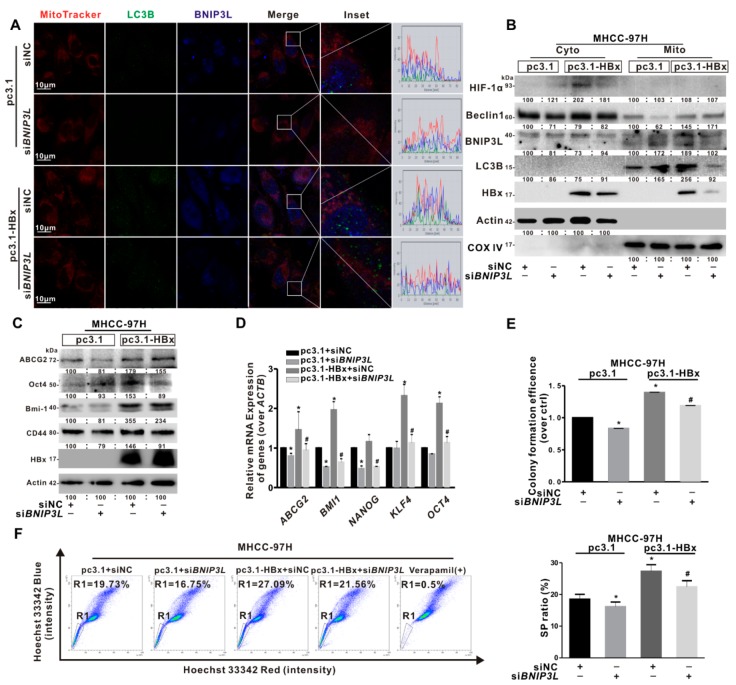
si*BNIP3L* inhibited cancer stemness induced by HBx-expressing in MHCC-97H cells. MHCC-97H cells without or with HBx-expressing were transiently transfected with pcDNA3.1 or pcDNA3.1-HBx (1 μg/mL) and si*BNIP3L* or siNC (50 nmol/L). (**A**) Representative images of the immunofluorescence co-staining for MitoTracker (red), BNIP3L (blue), and LC3B (green). The profiles of representative lines trace the intensities of fluorescence signals. Fluorescence curves with line intensity profile generated by Zen 2012 software were shown. Scale bar represents 10 μm. (**B**) The BNIP3L-dependent mitophagy-related proteins in cytoplasmic (Cyto) and mitochondrial (Mito) fractions. (**C**) The expression levels of cancer stemness-related proteins. The gray value of band was assessed by image-pro plus 6.0. The relative expression level was shown. (**D**) The mRNA levels of cancer stemness-related genes. The target gene transcription was normalized to *ACTB*. (**E**) The self-renewal capacity was measured by colony formation assay. (**F**) Percentage of the sorted SP cells (R1 gate) were detected by FCM (Left). Quantitative results were shown in bar graph (Right). SP: side population. R1 gate represented SP cells. * *P* < 0.05 as compared with pc3.1 group. ^#^
*P* < 0.05 as compared with pc3.1-HBx group. pc3.1: pcDNA3.1 transfection without HBx-expressing. pc3.1-HBx: pcDNA3.1-HBx transfection with HBx-expressing.

**Figure 7 cancers-12-00655-f007:**
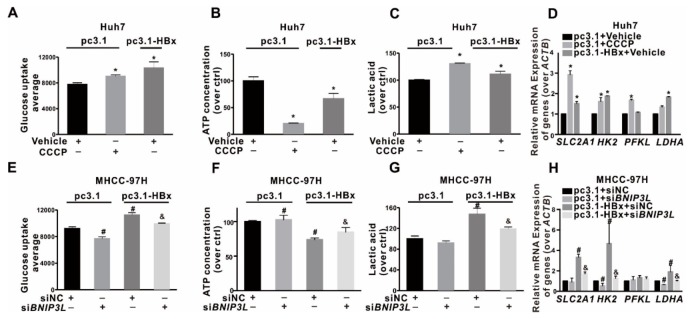
Relationship between the BNIP3L-dependent mitophagy and glycolysis metabolism reprogramming in HBx-expressing HCC cells. (**A**–**D**) Huh7 cells without or with HBx-expressing were transiently transfected with pcDNA3.1 or pcDNA3.1-HBx (1 μg/mL) for 8 h, and pretreated with CCCP (20 μM) for 3 h or not. (**A**) Glucose transport activity was evaluated by FCM. The levels of the intracellular ATP content (**B**), the extracellular lactic acid secretion (**C**), and the mRNA levels of glycolysis-related genes (**D**) were detected. (**E**–**H**) MHCC-97H cells without or with HBx-expressing were simultaneously transfected with si*BNIP3L* or siNC (50 nmol/L) for 8 h, and followed by restored culture for another 24 h. (**E**) Glucose transport activity was evaluated by FCM. The levels of the intracellular ATP content (**F**), the extracellular lactic acid secretion (**G**), and the mRNA levels of glycolysis-related genes (**H**) were detected. The target gene transcription was normalized to *ACTB. * P* < 0.05 as compared with pc3.1+Vehicle group, *^#^ P* < 0.05 as compared with pc3.1+siNC group, ^&^
*P* < 0.05 as compared with pc3.1-HBx+siNC group. pc3.1: pcDNA3.1 transfection without HBx-expressing. pc3.1-HBx: pcDNA3.1-HBx transfection with HBx-expressing.

**Figure 8 cancers-12-00655-f008:**
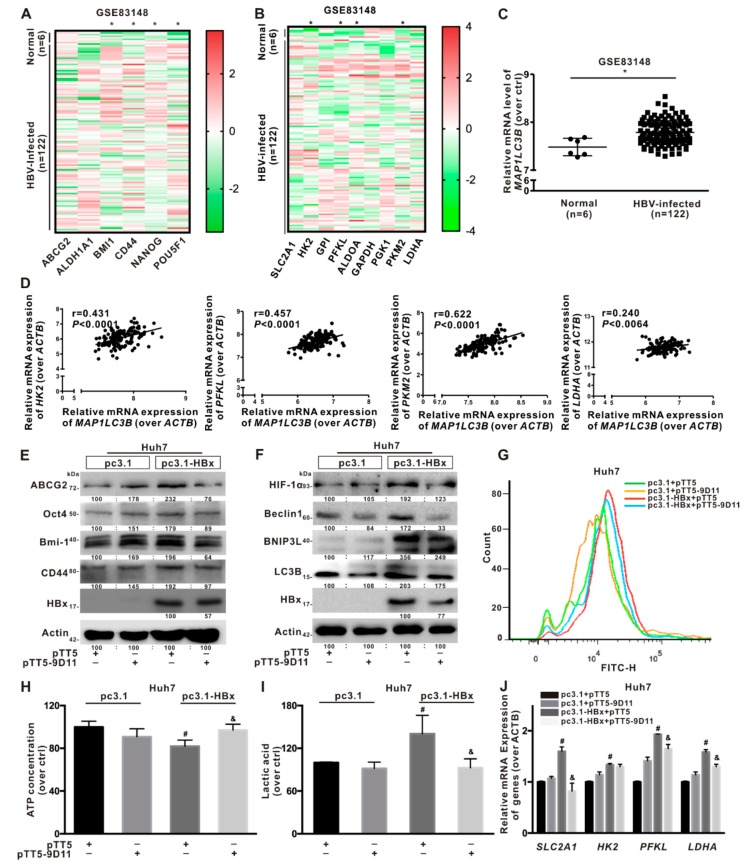
Anti-HBx targeting intervention to intracellular HBx inhibited the hepatocarcinogenesis associated with BNIP3L-dependent mitophagy. (**A**–**D**) The relative mRNA levels of indicated genes were obtained from NCBI, GEO database (GSE83148). The clinical cohort samples were derived from HBV-infected liver tissues (*n* = 122) and normal liver tissues (*n* = 6). The heat-map of the relative mRNA levels of cancer stemness-related genes (**A**) and glycolysis-related metabolism genes (**B**), the relative mRNA levels of *MAP1LC3B* gene (**C**), and the linear correlation of *MAP1LC3B* with glycolysis-related metabolism genes (**D**) were shown. (**E**–**J**) Huh7 cells without or with HBx-expressing were transiently transfected with pcDNA3.1 or pcDNA3.1-HBx (1 μg/mL), and then transiently transfected with pTT5 or pTT5-9D11 plasmids (200 ng/mL) for 8 h, and cultured for another 24 h. pTT5-9D11 plasmids encoded anti-HBx, a monoclonal antibody (mcAb), directed against intracellular HBx. (**E**) The expression levels of cancer stemness-related proteins. (**F**) The expression levels of BNIP3L-dependent mitophagy-related proteins. The gray value of band was assessed by image-pro plus 6.0. The relative expression level was shown. (**G**) Glucose transport activity was evaluated by FCM. The levels of the intracellular ATP content (**H**), the extracellular lactic acid secretion (**I**), and the mRNA levels of glycolysis-related genes (**J**) were detected. The target gene transcription was normalized to *ACTB*. * *P* < 0.05 as compared with normal liver tissues group. ^#^
*P* < 0.05 as compared with Huh7 group. ^&^
*P* < 0.05 as compared with HBx-expressing Huh7 group. pc3.1: pcDNA3.1 transfection without HBx-expressing. pc3.1-HBx: pcDNA3.1-HBx transfection with HBx-expressing.

**Figure 9 cancers-12-00655-f009:**
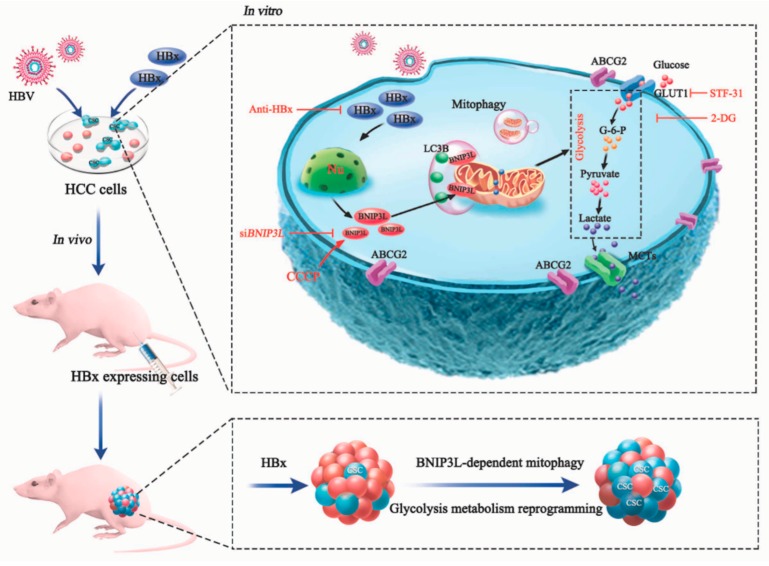
The scheme for molecular mechanisms of BNIP3L-dependent mitophagy-mediated reprogramming of glycolytic metabolism promoted cancer stemness in HBx-expressing HCC cells. In vivo, HBx-expressing promoted xenografted tumor growth, upregulated the expression of liver cancer stemness and BNIP3L-dependent mitophagy-related proteins, and increased the glycolytic metabolism. In vitro, HBx upregulated glycolysis metabolism reprogramming through BNIP3L-dependent mitophagy, and consequently enhanced the hepatic cancer stemness phenotypes in multiple HBx-expressing HCC cell models.

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
