# Peer review of "BNIP3L-Dependent Mitophagy Promotes HBx-Induced Cancer Stemness of Hepatocellular Carcinoma Cells via Glycolysis Metabolism Reprogramming"

_cancers, 2020, doi:10.3390/cancers12030655_

Round 1
Reviewer 1 Report
The article convincingly shows that BNIP3L-dependent mitophagy played an important role in maintaining cancer stemness in HBx-expressing HCC cells.
The first part of the article is devoted to proving previously known facts in selected experimental models:
-relation HBx with cancer stemness-related genes;
- HBx promoted cancer stemness phenotypes in HBV-related HBx-expressing HCC cell lines;
- glycolytic metabolism was reprogrammed in the HBx-expressing HCC cells and liver cancer stem cells (LCSCs).
Results showed that anti-HBx could reduce the hepatocarcinogenesis associated with HBx-induced BNIP3L-dependent mitophagy and glycolysis metabolism reprogramming in HCC cells.
However, the article is read very hard because contains a large number of figures (9 blocks in the main text and 6 in Supplementary Materials), and many abbreviations.
Remarks:
Title - line 2 –denpendent - replace with dependent;
Literal repetition of methods, for example lines 140-141 and 572-573;
In the text of the article there is no link to the figure 9;
Edit page No. In Supplementary Materials.
Author Response
Response to Reviewer 1 Comments
Dear Reviewer:
We would like to thank Cancers for giving us the opportunity to revise our manuscript “BNIP3L-dependent mitophagy promotes HBx-induced cancer stemness of hepatocellular carcinoma cells via glycolysis metabolism reprogramming”. We appreciate the thoughtful comments and valuable recommendations from you. We have carefully taken these suggestions into consideration in preparing our revision and made corrections throughout the manuscript. We provided point-by-point response to reviewers’ comments in below. Please note that the reviewers original comments are in black and our responses are in red.
Great thanks to you and the reviewers for the time and effort you expended on this paper.
Best wishes,
Zhong-Ning Lin, Ph.D; Yu-Chun Lin, Ph.D.
linzhn@xmu.edu.cn; linych@xmu,edu,cn
State Key Laboratory of Molecular Vaccinology and Molecular Diagnostics,
School of Public Health, Xiamen University,
Xiang’an South Road, Xiamen, Fujian 361102, PR China.
Title - line 2 –denpendent - replace with dependent;
Response: Thanks to the reviewers' suggestions, we have replaced denpendent with dependent. As follows: “BNIP3L-dependent mitophagy promotes HBx-induced cancer stemness of hepatocellular carcinoma cells via glycolysis metabolism reprogramming”
Literal repetition of methods, for example lines 140-141 and 572-573;
Response: We thank Reviewer for the generous and supportive comments.
In the original manuscript, Literal repetition of methods, lines 140-141 and 572-573: “The transfected cells were subcutaneously inoculated into the right flank of each mouse, and the tumors were allowed to grow for twenty-four days.”
In order to facilitate the reader's understanding, we further modify the language expression through English editing. We change “Constructing HCC xenograft tumors model with or without HBx-expressing, pcDNA3.1-HA-HBx or pcDNA3.1-HA plasmid were used to transfect Huh7 and MHCC-97H cells. The transfected cells were subcutaneously inoculated into the right flank of each mouse, and the tumors were allowed to grow for twenty-four days. As shown in Figure 1A, the tumors formed in HBx-expressing xenograft tumors (HBx-expressing group) were significantly rapider than those transfected with pcDNA3.1-HA xenograft tumors (control group). Compared with control group, HBx-expressing group significantly increased the size and weight of tumors (Figure 1B, C).” (In the original manuscript Page 3, lines 138-144) to “To evaluate the effect of HBx-expressing on tumor growth in vivo was examined using mouse subcutaneous xenograft model. Tumor xenografts derived from HBx-expressing Huh7 and MHCC-97H cells exhibited the tumors formed rapider, bigger volumes and higher weights than those from empty vector (pcDNA3.1-HA plasmid) -transduced cells (Figure 1A, B, C).” (In the revised manuscript Page 4, lines 223-226).
In the text of the article there is no link to the figure 9;
Response: Thanks for your kind suggestion. We added Figure 9 in the section 5. Conclusions. (Revised manuscript Page 20, Line 738).
Edit page No. In Supplementary Materials.
Response: Thanks for your kind suggestion. We have edited the page No. In Supplementary Materials. (Revised Supplementary Materials Page 1-17).

Reviewer 2 Report
In this study, Chen et al investigated how HBV x protein (HBx) mediates the induction and maintenance of cancer stemness in HBV-related hepatocellular carcinoma (HCC) cells. Using HCC cell xenograft model and HCC cell lines, the authors showed that HBx promotes tumor growth and glycolytic metabolism both in vivo and in vitro. The authors further showed that BNIP3L-dependent mitophagy might be involved in the regulation of cancer stemness an glycolytic metabolism. Finally, the authros showed that intervention to intracellular HBx by anti-HBx antibody inhibited the BNIP3L-dependent mitophagy and glycolysis metabolism, further emphysizing the importantce of HBx protein in the pathogenesis.
This is a very interesting study, proposing a mechanism of how HBx protein might promoter tubor growth of HCC. However, the manuscript in its current format is lack of proof-reading and logic connection between important pieces of data. Significant effort is needed to clarify the rationals for the experiments and to make the story easy to understand.
Specific points:
The authors wanted to understand how HBx might induce cancer stemness in HBV-related HCC. It might be more convincing if the authors could test some of the key experiments in primary human hepatocytes, such as HBx induced glycolytic metabolism (Figure 3) and BNIP3L-dependent mitophagy (Figure 5)? In the introduction section, the authors introduced two studies leading to different functions of HBx protein in mitochondria dynamics and/or HCC development. The authors then mentioned that HBx interacts with HIF-1a, which upregulates BNIP3L transcription in breast tumors. How this info leads to the question “there is a need for detailed analysis of BNIP3L-dependent mitophage in HBx-expressing HCC…”? The authors need to give more detailed backgroup introduction of BNIP3L and mitophagy. In the xenograft model (Figure 1), the authors observed that tumor size and weight are significantly increased in HBx-expressing group. Do the HBx-expressing HCC cells have impact on their neighbour cells that do not express HBx? The authors noticed that the expression of many stemness-related genes were upregulated at their transcript and/or protein levels. Does the HBx have impact on the overall percetage of cancer stem cell (CSC)? What did the author mean by “liver CSCs were enriched in Huh7 cells by sphere-formation assy (line 157-158)”? Similar to the question mentioned previously, does expression of HBx increase the overall percentage of CSC in HCC lines? Or HBx mainly increases the expression of stemness-related genes, but not the CSC population? For the SP cells, in Huh7 cells, HBx increases their ratio from 1.43% to 1.73%; does this have biological meanings? What did the author mean here “It was suggested glycolytic metabolism was reprogrammed in LCSCs (line 184-185)”? Was this a conlcusion from previous studies (need references here) or a conlcusion from data showing in the manuscript? The authors analysed some deposited data-sets and presented their analysis in Figure 8. It’s nice to validate some of the key findings from this study bioinformatically. However, these datasets were mainly dealing with chronic HBV-infected patients, as compared to normal liver tissue. The authors should make it clear in the result section that this is not to emphasize the imporatance of HBx protein, but infection. Sicen numerous studies have shown that HBx protein plays important roles in HCC development and progression, by manipulating cell cycle, cell proliferation, and survival. Therefore, in the discussion section, it might be better if the authors could futher discuss the findings from the current study by comparing them to previous studies.Author Response
Response to Reviewer 2 Comments
Dear Reviewer:
We would like to thank Cancers for giving us the opportunity to revise our manuscript “BNIP3L-dependent mitophagy promotes HBx-induced cancer stemness of hepatocellular carcinoma cells via glycolysis metabolism reprogramming”. We appreciate the thoughtful comments and valuable recommendations from you. We have carefully taken these suggestions into consideration in preparing our revision and made corrections throughout the manuscript. We provided point-by-point response to reviewers’ comments in below. Please note that the reviewers original comments are in black and our responses are in red.
Great thanks to you and the reviewers for the time and effort you expended on this paper.
Best wishes,
Zhong-Ning Lin, Ph.D; Yu-Chun Lin, Ph.D.
linzhn@xmu.edu.cn; linych@xmu,edu,cn
State Key Laboratory of Molecular Vaccinology and Molecular Diagnostics,
School of Public Health, Xiamen University,
Xiang’an South Road, Xiamen, Fujian 361102, PR China.
The authors wanted to understand how HBx might induce cancer stemness in HBV-related HCC. It might be more convincing if the authors could test some of the key experiments in primary human hepatocytes, such as HBx induced glycolytic metabolism (Figure 3) and BNIP3L-dependent mitophagy (Figure 5)?
Response 1: We appreciate the comments from the reviewer.
To explore the molecular mechanisms by which HBx promoted CSCs generation in HBV-related HCC. We agreed with the suggestions from the reviewer, “It might be more convincing if the authors could test some of the key experiments in primary human hepatocytes.” The evidences for supporting the issue were based on the relevant literature and our previous report in the field of HBV infection-induced liver diseases as following:
Casciano et al reported that HBx modulates mitochondrial calcium signaling to stimulate HBV replication mediated mitochondrial metabolism and mitochondrial quality control (MQC) disorder in primary human hepatocytes, similar to HBx effects in HepG2 cells and primary rat hepatocytes[1,2]. In recent years, the human liver chimeric mouse (HLC) with primary human hepatocytes (PHHs) engraftment has been demonstrated to be a useful animal model to study hepatitis B virus (HBV) pathogenesis [3,4]. Based on the HLC mouse model, our previous report further found that HBx exposure affects metabolic changes in the primary human hepatocytes (PHHs) from HBV-infected HLC ex vivo, the liver of HBV-Tg mice in vivo, as well as multiple HBx-expressing human hepatic cell lines in vitro. Our results suggested that HBx triggers metabolic abnormalities in liver cell through interfering with mitochondrial dynamics [5].
In the current study, multiple HBx-expressing cell models, side population (SP) of ATP-binding cassette sub-family G member 2 (ABCG2) positive subset, or sphere-forming cells with stem-like phenotypes were established. For the studies of mechanism, we proposed a positive feedback loop, in which HBx induced BNIP3L-dependent mitophagy (Figure 5) which upregulated glycolytic metabolism (Figure 3), increasing cancer stemness of HCC cells in vivo and in vitro (Figure 9). In the future, based on the HLC mouse model, we will further explore the molecular mechanisms by which HBx promoted CSCs generation in HBV-related HCC, which will help us have a deeper understanding of HBV-related HCC.
- Casciano J C, Bouchard M J. Measuring Changes in Cytosolic Calcium Levels in HBV-and HBx-Expressing Cultured Primary Hepatocytes[M]//Hepatitis B Virus. Humana Press, New York, NY, 2017: 143-155.
- Casciano J C, Bouchard M J. Hepatitis B virus X protein modulates cytosolic Ca2+ signaling in primary human hepatocytes[J]. Virus research, 2018, 246: 23-27.
- Douglas DN, Kneteman NM. Mice with Chimeric Human Livers and Their Applications. Methods Mol Biol. 2019;1911:459–479.
- Yuan L, Liu X, Zhang L, et al. Optimized HepaRG is a suitable cell source to generate the human liver chimeric mouse model for the chronic hepatitis B virus infection[J]. Emerging microbes & infections, 2018, 7(1): 1-17.
- Chen Y Y, Lin Y, Han P Y, et al. HBx combined with AFB1 triggers hepatic steatosis via COX‐2‐mediated necrosome formation and mitochondrial dynamics disorder[J]. Journal of cellular and molecular medicine, 2019, 23(9): 5920-5933.
In the introduction section, the authors introduced two studies leading to different functions of HBx protein in mitochondria dynamics and/or HCC development. The authors then mentioned that HBx interacts with HIF-1a, which upregulates BNIP3L transcription in breast tumors. How this info leads to the question “there is a need for detailed analysis of BNIP3L-dependent mitophage in HBx-expressing HCC…”? The authors need to give more detailed backgroup introduction of BNIP3L and mitophagy.
Response 2: We thank the reviewer for the generous and supportive comments.
We apologize for the lack of detailed backgroup introduction of BNIP3L and mitophagy in our previous manuscript. Based on the reviewer's suggestions, to better introduce the role of BNIP3L and mitophagy in HBx-expressing HCC, we have supplemented some related issues in the second paragraph of the section 1. Introduction (In the revised manuscript Page 2-3). Please note that the original manuscript are in black and our modifications are in red. As shown below:
HBV infection-triggered alteration of mitochondrial metabolism and mitochondrial quality control (MQC) has recently become a hot topic in cancer research[8]. Mitophagy, one of the critical component of MQC, is a degradation process that specifically targets impaired mitochondria to autophagosomes for the maintaining of mitochondria homeostasis. In recent years, the molecular mechanism of mitophagy has been extensively studied, mitophagy regulatory pathways are classified as PTEN-induced kinase 1 (PINK1)/Parkin-dependent, mitochondrial receptors–dependent such as B-cell lymphoma 2/adenovirus E1B interacting 19 kDa-interacting protein 3 (BNIP3), BNIP3-like (BNIP3L), and FUN14 domain containing 1 (FUNDC1), and lipid-mediated [9, 10]. PINK1/Parkin-dependent mitophagy might play a double-faceted role in hepatic cancer cells depending on different cellular context[11]. On the one hand, HBx induced aberrant mitochondria dynamics and promoted PINK1/Parkin-dependent mitophagy, to promote cell survival and possibly viral persistence[12]. On the other hand, It was suggested that thyriod hormone eliminated HBx-targeting mitochondria via PINK1/Parkin pathway in hepatocytes, and consequently prevented HBx-induced HCC[13]. It was reported that HBx directly interacted with the basic helix–loop–helix (bHLH) domain of hypoxia inducible factor-1α (HIF-1α) to increase its transcriptional activity and protein level[14]. Interestingly, HIF-1 target gene BNIP3L has been reported to cause mitochondrial dysfunction and cell death in breast tumors[15, 16]. BNIP3L at the outer mitochondrial membrane interacts with processed LC3 at phagophore membranes to promote the occurrence of mitophagy. It was considered to be important for mitochondrial clearance during reticulocyte maturation, and also participated in mitophagy in an energy-dependent manner, as well as mataining the stemness[17, 18]. Importantly, mitophagy acts as a key mechanism for developing and maintaining stemness. During chemotherapy, colorectal CSCs activated BNIP3L-dependent mitophagy to clear the damaged mitochondria and maintain cell survival[19]. However, whether HBx could induce BNIP3L-dependent mitophagy in the progression of HBV-related HCC remains to be elucidated. Therefore, more detailed experimental investigation underlying the role of mitophagy in the acquisition and maintenance of stemness in HBV-related HCC is worthy of further study.
9. Wu, H.; Chen, Q. Hypoxia activation of mitophagy and its role in disease pathogenesis. Antioxid Redox Signal 2015, 22, 1032-1046.
10.Chu, C.T.; Ji, J.; Dagda, R.K.; Jiang, J.F.; Tyurina, Y.Y.; Kapralov, A.A.; Tyurin, V.A.; Yanamala, N.; Shrivastava, I.H.; Mohammadyani, D., et al. Cardiolipin externalization to the outer mitochondrial membrane acts as an elimination signal for mitophagy in neuronal cells. Nat Cell Biol 2013, 15, 1197-1205.
17.Sandoval, H.; Thiagarajan, P.; Dasgupta, S.K.; Schumacher, A.; Prchal, J.T.; Chen, M.; Wang, J. Essential role for Nix in autophagic maturation of erythroid cells. Nature 2008, 454, 232-235.
In the xenograft model (Figure 1), the authors observed that tumor size and weight are significantly increased in HBx-expressing group. Do the HBx-expressing HCC cells have impact on their neighbour cells that do not express HBx?
Response 3: Thanks for your kind suggestion.
In the xenograft model (Figure 1), to evaluate the effect of HBx-expressing on tumor growth in vivo was examined using mouse subcutaneous xenograft model. We observed that tumor xenografts derived from HBx-expressing Huh7 and MHCC-97H cells exhibited the tumors bigger size and higher weights than those control group (Revised manuscript Figure 1B, C) . Our results suggested that HBx promotes growth of hepatocellular carcinoma cells.
Numerous studies have indicated that HBx protein encoded by the HBV virus X gene has been considered to be oncogenic and implicated in hepatocarcinogenesis [1-3]. It has HBx plays important roles in HCC development and progression via regulating a series of biological processes inducing the malignant transformation of liver cells. Zhu et al showed that HBx through stimulating expression of AFP to promote malignant behaviors of human normal liver cells and HCC cells [4]. In transgenic mice, HBx was shown to induce HCC and chemical carcinogen-induced liver cancer [5].
Cell-cell communication plays a critical role in a myriad of processes, such as angiogenesis and carcinogenesis, in multi-cellular organisms [6]. For example, recent studies highlighted the importance of exosomes in cell-to-cell communication [7]. Marked and specific changes in exosome protein contents were also detected by comparing exosomes secreted by HBx-infected Huh7 cells with those secreted by a control group [8]. The exosomal cargo secreted by HBx-expressing cells had a profound effect on the recipient hepatic cells including creation of a milieu conducive for cellular-transformation [9]. Thus, these study unfolds a novel role of HBx in intercellular communication by facilitating horizontal transfer of viral gene products and other host factors via exosomes in order to support viral spread and pathogenesis. Overall, the complexities of cell-cell communication and the possibilities for modulation provide new opportunities for treating cancers. Regarding “Do the HBx-expressing HCC cells have impact on their neighbour cells that do not express HBx?". We think the HBx-expressing HCC cells have impact on their neighbour cells that do not express HBx.
- Liu S, Koh SS, Lee CG. Hepatitis B Virus X Protein and Hepatocarcinogenesis. Int J Mol Sci. 2016;17(6):940. Published 2016 Jun 14.
- Niu Y, Xu M, Slagle BL, et al. Farnesoid X receptor ablation sensitizes mice to hepatitis b virus X protein-induced hepatocarcinogenesis. Hepatology. 2017;65(3):893–906.
- Mao X, Tey SK, Ko FCF, et al. C-terminal truncated HBx protein activates caveolin-1/LRP6/β-catenin/FRMD5 axis in promoting hepatocarcinogenesis. Cancer Lett. 2019;444:60–69.
- Zhu M, Lu Y, Li W, et al. Hepatitis B Virus X Protein Driven Alpha Fetoprotein Expression to Promote Malignant Behaviors of Normal Liver Cells and Hepatoma Cells. J Cancer. 2016;7(8):935–946. Published 2016 May 12. doi:10.7150/jca.13628
- Ahodantin J, Bou-Nader M, Cordier C, et al. Hepatitis B virus X protein promotes DNA damage propagation through disruption of liver polyploidization and enhances hepatocellular carcinoma initiation. Oncogene. 2019;38(14):2645–2657.
- Hughes BR, Mirbagheri M, Waldman SD, Hwang DK. Direct cell-cell communication with three-dimensional cell morphology on wrinkled microposts. Acta Biomater. 2018;78:89–97.
- Xu R, Rai A, Chen M, Suwakulsiri W, Greening DW, Simpson RJ. Extracellular vesicles in cancer - implications for future improvements in cancer care. Nat Rev Clin Oncol. 2018;15(10):617–638.
- Zhao X, Wu Y, Duan J, et al. Quantitative proteomic analysis of exosome protein content changes induced by hepatitis B virus in Huh‐7 cells using SILAC labeling and LC‐MS/MS. J Proteome Res. 2014; 13: 5391–402.
- Kapoor N R, Chadha R, Kumar S, et al. The HBx gene of hepatitis B virus can influence hepatic microenvironment via exosomes by transferring its mRNA and protein[J]. Virus research, 2017, 240: 166-174.
What did the author mean by “liver CSCs were enriched in Huh7 cells by sphere-formation assy (line 157-158)”? Similar to the question mentioned previously, does expression of HBx increase the overall percentage of CSC in HCC lines? Or HBx mainly increases the expression of stemness-related genes, but not the CSC population? For the SP cells, in Huh7 cells, HBx increases their ratio from 1.43% to 1.73%; does this have biological meanings?
Response 4: We appreciate the comments from the reviewer.
To make it easier for the reader to understand “liver CSCs were enriched in Huh7 cells by sphere-formation assy (In the original manuscript Page 4, line 157-158)”, We have modified it as follows:“To further characterize stemness features of HCC cells, Huh7 cells were cultured in the ultra-low attachment plate for two weeks resulted in the enrichment of liver CSCs (LCSCs) by using sphere-formation assay.”
In the current study, to explore the role of HBx-expressing in cancer stemness, two HCC cell lines were transiently transfected with pcDNA3.1-HBx to establish a HBx-expressing cell model. Our result found that the mRNA levels of cancer stemness-related genes, including ABCG2, BMI1, NANOG, KLF4, and OCT4 were increased, and the protein expression of ABCG2, Oct4, Bmi-1, and CD44 were also increased in HBx-expressing HCC cells (Revised manuscript Figure 2A, B). Besides, HBx increase the overall percentage of CSC in HCC lines (Revised manuscript Figure 2C). Our results suggested that HBx promoted cancer stemness phenotypes in HBV-related HBx-expressing HCC cell lines.
Consistent with our research, Wang et al. also found that HBx not only promoted SP cells ratios increased from 1.09% and 1.35% of the total SMMC-7721 and Huh7 cells in the control group to 2.44% and 3.22% in the HBx-expressing group, but also increases the expression of stemness-related genes [1].
Tumors are composed of non-homogeneous cell populations exhibiting varying degrees of genetic and functional heterogeneity. CSCs are a small subset of heterogeneous cell populations which have phenotypes of self-renewal [2]. Our results need to be further validated in preclinical models to provide clues for CSC-based treatment strategies. The results of our and other laboratories need further validation in preclinical models to provide clues for CSC-targeting therapies in various cancers [3].
- C. Wang, M.D. Wang, P. Cheng, H. Huang, W. Dong, W.W. Zhang, P.P. Li, C. Lin, Z.Y. Pan, M.C. Wu, W.P. Zhou, Hepatitis B virus X protein promotes the stem-like properties of OV6+ cancer cells in hepatocellular carcinoma, Cell death & disease, 8 (2017) e2560.
- Muramatsu S, Tanaka S, Mogushi K, Adikrisna R, Aihara A, Ban D, et al. Visualization of stem cell features in human hepatocellular carcinoma reveals in vivo significance of tumor-host interaction and clinical course. Hepatology. 2013;58:218-28.
- Saygin C, Matei D, Majeti R, Reizes O, Lathia JD. Targeting Cancer Stemness in the Clinic: From Hype to Hope. Cell Stem Cell. 2019;24(1):25–40. doi:10.1016/j.stem.2018.11.017
What did the author mean here “It was suggested glycolytic metabolism was reprogrammed in LCSCs (line 184-185)”? Was this a conlcusion from previous studies (need references here) or a conlcusion from data showing in the manuscript?
Response 5: We thank the reviewer for the generous and supportive comments.
In the section 2.1.3. Glycolytic metabolism was reprogrammed in the HBx-expressing HCC cells and LCSCs. To evaluate the role of HBx-expressing on glycolytic metabolism and the metabolic pattern in vitro, we performed sphere-formation assay in Huh7 cells. We found that the glucose uptake was higher in sphere-formed LCSCs than parental Huh7 cells by FCM (Revised manuscript Figure 3A). And the content of intracellular ATP was decreased and the secretion of extracellular lactic acid was increased in sphere-formed LCSCs of Huh7 cells (Revised manuscript Figure 3B, C). In addition, the mRNA levels of glycolysis-related genes, including SLC2A1, HK2, PFKL, and LDHA were significantly higher, whereas the OXPHOS-related genes mRNA levels of CytB in Complex III, and ATP6, ATP8 in Complex V were lower in sphere-formed LCSCs than those in parental Huh7 cells (Revised manuscript Figure 3D, E).
Based on the above experiments, the overall results reflect a conlcusion showing in the manuscript “It was suggested glycolytic metabolism was reprogrammed in LCSCs (In the original manuscript line 184-185)”. Considering the questions raised by the reviewer, we would like to modify “It was suggested glycolytic metabolism was reprogrammed in LCSCs”. The conlcusion description of the revised manuscript now is “The above results suggested that glycolytic metabolism was reprogrammed in LCSCs (In the revised manuscript)”.
The authors analysed some deposited data-sets and presented their analysis in Figure 8. It’s nice to validate some of the key findings from this study bioinformatically. However, these datasets were mainly dealing with chronic HBV-infected patients, as compared to normal liver tissue. The authors should make it clear in the result section that this is not to emphasize the imporatance of HBx protein, but infection.
Response 6: We thank the reviewer for the generous and supportive comments.
In Figure 8, To confirm whether our in vitro and in vivo findings that BNIP3L-denpendent mitophagy promotes HBx-induced cancer stemness of HCC cells via glycolysis metabolism reprogramming in chronic HBV-infected patients. In a clinical cohort of chronic HBV-infected patients, We further examined the relative mRNA levels of cancer stemness-related genes (such as BMI1, CD44, NANOG, and POU5F1), glycolysis-related genes (including HK2, PFKL, PKM2, and LDHA), and mitophagy-related genes. These datas indicated that HBV-infection may associate with the induction of cancer stemness phenotype, glycolytic metabolism, and mitophagy in liver, where mitophagy would positively correlated with glycolytic metabolism in hepatocytes. To verify the role of mitophagy and glycolysis metabolism reprogramming pathways at the HBV-infection population level. Therefore, we agree with the opinions of the review experts, these datasets were mainly confirming our in vitro and in vivo findings in chronic HBV-infected patients. Besides, in the discussion section, We futher discuss the findings from the current study by comparing them to previous studies (In the revised manuscript).

Round 2
Reviewer 2 Report
The authors have thoroughly and satisfactorily addressed my previous points/concerns and I have no further comments.